

# The relationship between appearance anxiety and depression among students in a medical university in China: a serial multiple mediation model

Xiaobing Xian[1],*, Tengfei Niu[2],*, Yu Zhang[1], Xilin Zhou[1], Xinxin Wang[3], Xin Du[4], Linhan Qu[5], Binyi Mao[3], Ying He[4], Xiyu Chen[6] and Mengliang Ye[1]

[1] School of Public Health, Chongqing Medical University, Chongqing, China
[2] Department of Basic Courses, Chongqing Medical and Pharmaceutical College, Chongqing, China
[3] College of Traditional Chinese Medicine, Chongqing Medical University, Chongqing, China
[4] The Second Clinical College, Chongqing Medical University, Chongqing, China
[5] The First Clinical College, Chongqing Medical University, Chongqing, China
[6] College of Pediatrics, Chongqing Medical University, Chongqing, China
* These authors contributed equally to this work.

Corresponding author
Mengliang Ye,
yemengliang@cqmu.edu.cn

## ABSTRACT

**Background:** Appearance anxiety and depression have become common and global public health problems worldwide, especially among adolescents. However, few studies have revealed the mechanisms between them. This study aimed to explore the multiple mediating roles of interpersonal sensitivity and social support between appearance anxiety and depression among medical college students.

**Methods:** With 13 invalid samples excluded, 724 college students participated in our survey and completed questionnaires. The average age of 724 samples was 19.8 ± 2.02 including freshman to senior year and graduate school and above; 31.9% of the participants were male and 68.1% were female. SPSS 25.0 and Hayes' PROCESS macro were used for statistical description, correlation analysis and built multiple mediation models.

**Results:** Appearance anxiety can not only directly affect depression, but also indirectly affect depression through three significant mediating pathways: (1) IS (B = 0.106, 95% CI [0.082–0.132]), which accounted for 49.77% of the total effect, (2) SS (B = 0.018, 95% CI [0.008–0.031]), which accounted for 8.45% of the total effect, and (3) IS and SS (B = 0.008, 95% CI [0.003–0.014]), which accounted for 3.76% of the total effect. And the total mediating effect was 61.97%.

**Limitations:** It is a cross-sectional research method and the causal relationship is unclear.

**Conclusions:** This study found that lower interpersonal sensitivity and higher social support can effectively reduce depression caused by appearance anxiety among college students. The schools and relevant departments should take measures to reduce the interpersonal sensitivity of college students and establish reliable social support, so as to reduce the occurrence of depression.

# INTRODUCTION

Depression is an important public health problem that affects more than 300 million people worldwide. A study from *Wang et al. (2023)* has demonstrated that about 22% of college students in China suffer from depression, and the prevalence of depression among Chinese medical students is 27% (*Jin et al., 2022*). So this shows that college students have an increasing risk of depression, the degree of which is higher than students in middle and elementary school, especially among medical college students (*Wang et al., 2023*). Depression and anxiety are multifactorial and complex psychological problems with possible risk factors including psychological, academic, biological, lifestyle, social, and economic. In psychology, for example, students with higher neurotic and more introverted are more likely to be depressed from psychological view. Multiple college-related poor lifestyles such as smoking, drinking, and substance abuse also increase the risk of depression among college students. Having a supportive social network, like family and friendships, can also influence a student's mood and reduce the likelihood of depression. If suffering from depression, students, especially medical students, are more likely to give up solving complex problems and feel confused during their studies (*Turana et al., 2022*; *Rengasamy et al., 2021*; *Mofatteh, 2021*). These behaviors may lead to hypertension, obesity, diabetes, and other diseases in the future, which seriously reduce the quality of life and increase the economic burden (*Petersen et al., 2022*). More importantly, the global suicide rate caused by depression is increasing every year and depression becomes a major risk factor for adolescents' suicide (*Turana et al., 2022*). According to existing studies, although there are many factors influence adolescents' depression, such as anxiety, appearance satisfaction, interpersonal sensitivity, and social support, the complex relationships among the influencing factors are still unclear and need to be further investigated (*Kim, Han & Won, 2023*; *Xu et al., 2022*; *Bi & Wang, 2022*; *Zvolensky et al., 2021*). Adolescence is a time when individuals go through major physical and psychological changes as they grow up, during which adolescents pay special attention to their image and are susceptible to the influence of the mass media's aesthetic standards for appearance and figure. If they realize that they do not meet public social standards, they are prone to appearance anxiety, which is likely to lead to further depression (*Lim & Kwon, 2023*).

Appearance anxiety (AA) refers to a condition when an individual is overly concerned about their appearance. They may become unconfident and anxious about their appearance when they perceive social standards or poor evaluations of themselves from others (*Gao et al., 2023*). With the increasing rate of science and technology development, various social platforms and media have become flourishing and have been invading in college students' lives (*de Vries et al., 2016*). They gradually begin to pay excessive attention to their appearance and worry about others' evaluation of their appearance. These may even trigger their anxiety (*Marengo et al., 2018*). Some studies have pointed out that adolescence is a time in human development when they go through drastic physical and

psychological changes. During adolescence, students pay more attention to their looks and are always unconsciously exposed to unrealistic standards of beauty, appearance, or body type from the mass media. Their self-esteem probably declines if they think that they do not meet these standards, resulting in anxiety and depression (*Hai & Yang, 2022*). Appearance influences the formation and establishment of self-esteem in adolescence and young adulthood. Anxiety caused by appearance probably makes them negative due to a lack of confidence, which leads to a decrease in the level of self-esteem of the individual (*Lim & Kwon, 2023*), resulting in an inability to integrate into society and to positively face the challenges of interpersonal relationships. Therefore, it is necessary to make a further study on the effect of AA on depression among medical college students in order to explore ways to reduce the rate of depression among them. Advances in a cognitive behavioral model of body dysmorphic disorder proposed that when individuals compare themselves to the least likely ideal appearance, they will be negative, which leads to self-protective behaviors such as avoidance (*Bilsky et al., 2022*; *Zimmer-Gembeck et al., 2021*). Some adolescents may be bullied, harassed, and ridiculed because of their appearance, which exacerbates adolescents' AA and gradually develops distorted perceptions and aversion to their own appearance (*Zimmer-Gembeck et al., 2021*; *Bucchianeri, Eisenberg & Neumark-Sztainer, 2013*). When adolescents compare themselves upward with others or compare their ideal selves with their real-life selves, a discrepancy is created, which can lead to emotional vulnerability, anxiety and exacerbate depression (*Higgins, 1987*; *Festinger, 1954*). Anabolic-androgenic steroid users are at greater risk of depression if they have defective appearances (*Griffiths et al., 2018*). Dissatisfaction with physical appearance can act as a mediator to influence the relationship between self-esteem and depression (*Kim, Han & Won, 2023*). Appearance influences the formation and establishment of self-esteem in adolescence and young adulthood. External anxiety can lead to a lack of confidence and negativity and further a decrease in an individual's level of self-esteem, resulting in an inability to integrate into society and positively face the challenges of interpersonal relationships (*Liao et al., 2023*). Hairi et al. found that a person's dissatisfaction with his or her appearance can lead to lowered self-esteem, which in turn can trigger depression (*Hai & Yang, 2022*). A study in Coimbatore showed that 77.6% of college women were dissatisfied with their body image and that depression was significantly associated with it (*Ganesan, Ravishankar & Ramalingam, 2018*). At the same time, anxiety and depression are closely related–they are comorbidity, and anxiety is a risk factor for depression (*Belzer & Schneier, 2004*). Studies have demonstrated that elevated levels of anxiety are associated with depression among college students in Latin America (*Zvolensky et al., 2021*). Based on existing studies, it can be reasonably speculated that medical college students' dissatisfaction with their appearance is likely to cause AA, which in turn increases the likelihood of depression.

Interpersonal sensitivity (IS) is one of the mental health problems faced by contemporary college students (*Ding et al., 2021*). IS was first proposed by *Boyce & Parker (1989)* and is considered as a personality trait that usually manifests itself as an over-understanding of the behavior and emotions of others (*Masillo et al., 2012*). Individuals sometimes pay too much attention to their own relationships and fear the

rejection or criticism of others in social interactions, which is also a symptom of IS (*Xu et al., 2022*). Discomfort and anxiety will happen in people with IS traits when they are interacting with people. Such discomfort leads to social anxiety and a strong sense of low esteem, making individuals vulnerable to developing depression. They also often change their behavior to conform to the expectations and ideas of others (*Boyce & Parker, 1989*; *Derogatis & Melisaratos, 1983*). Some researchers have used structural equation modeling to show that IS moderates anxiety states in participants who are abused as children, and that also moderates AA in adolescents (*Maftei, 2022*; *Nakazawa et al., 2021*). Among Chinese college students, negative emotion is related to IS and is one of the predictors of depression (*Xu et al., 2022*; *Ding et al., 2021*). Whether IS can be used as a mediating variable to moderate the relationship between AA and depression among college students needs to be validated through further research.

Social support (SS) refers to a social network of family, friends, teachers, and classmates that provides emotional support and practical help to individuals (*Cohen, 2004*). When encountering difficulties, college students can seek help from family, friends, *etc.*, so as to obtain solutions to problems, which greatly relieves physical and mental stress (*Xu et al., 2022*). Therefore, SS is a key factor in the mental health of college students and makes an important contribution to the mental health of adolescents (*Wang et al., 2022*; *DuBois et al., 1994*). *Peirce et al. (2000)* constructed a model based on social support theory and found that SS was negatively associated with depression. The study made by *Jaycox et al. (2009)* reported an effect of social interaction on depression in adolescents. SS can significantly reduce the anxiety levels of community residents during the COVID-19 outbreak in Turkey in 2020 (*Özmete & Pak, 2020*). Body image and SS in patients with Psoriasis were found to be major contributors to depression (*Wojtyna et al., 2017*). Meanwhile, SS plays an important role in reducing depressive symptoms and IS, and higher levels of SS can reduce the severity of an individual's depressive symptoms and IS (*Mei et al., 2022*).

Based on the available studies, it can be speculated that there is some relationship between AA, IS, SS, and depression. There are few studies on the mechanisms between AA and depression among Chinese college students, studying in a medical school, and there is a lack of studies demonstrating the mediating role of IS and SS in the relationship between AA and depression. Therefore, this study aimed to investigate the relationship between AA, IS, SS, and depression, as well as to explore the multiple mediating roles of IS and SS between AA and depression in medical college students. The purpose of this study is to reach a deeper understanding of the factors influencing depression and to provide a theoretical basis for the development of public health policies in relevant sectors. This is also to prevent depression at the source, consciously reduce the risk of depression among medical college students, and maintain their physical or mental health at a high level. Based on the above, the following three hypotheses are proposed:

H1: AA positively predicts depression among college students studying in a medical school.

H2: AA indirectly predicts depression through high IS among college students studying in a medical school.

H3: AA indirectly predicts depression through low SS among college students studying in a medical school.

H4: AA indirectly predicts depression through high IS and low SS among college students studying in a medical school.

## MATERIALS AND METHODS

### Participants and procedure

This study adopted a cross-sectional survey. From February 2nd to February 5th, 2023, a convenience sampling was used to recruit 737 college students for a questionnaire survey at Chongqing Medical University. Posters, applets, QR codes, and links were utilized to disseminate and recruit research subjects. The inclusion criteria for this study were students who were enrolled in Chongqing Medical University (including undergraduate, graduate, and doctoral students) and were willing to participate in this study. Respondents were briefed by professional investigators on the survey content and purpose prior to the survey to seek informed consent. If the subject was underage, investigators obtained informed consent from their guardians *via* the Internet in advance. The questionnaire included the basic demographic variables, the Appearance Anxiety Scale-Brief Version (AASBV), the Perceived Social Support Scale (PSSS), the SCL-90 Interpersonal Sensitivity Subscale, and the Patient Health Questionnaire (PHQ-9). The questionnaires were distributed and collected relying on the Questionnaire Star platform online. The inclusion criteria were that the results of questionnaires have no missing items, obvious logical errors, and invalid numbers (*e.g.*, the ones whose height is over 2.5 m or less than 1 m; the ones whose weight is less than 30 kg or more than 125 kg; the ones whose age are below 16 or bigger than 36). More precisely, all the participants were university students at Chongqing Medical University and after removing missing and invalid data, the remaining sample size was 724.

### Measurement

#### Appearance anxiety

Appearance anxiety was measured by the Appearance Anxiety Scale-Brief Version (AASBV) written by *Keelan, Dion & Dion (1992)*. The Appearance Anxiety Scale-Brief Version consists of 14 items. A 5-point Likert scale ranging from 1 (never) to 5 (almost always) is used, which reflects the respondents' combined attitudes toward appearance anxiety with a total score ranging from 14 to 70. The higher the overall score is, the higher the degree of appearance anxiety is (*Jin et al., 2022*). The chosen Chinese version of the Appearance Anxiety Scale is a short version of good reliability and validity and the Cronbach's alpha measured was 0.876.

### Social support

The Perceived Social Support Scale (PSSS) is used to measure social support (*Tonsing, Zimet & Tse, 2012*). Response to an item was measured on a 7-point Likert scale, ranging from 1 (strongly disagree) to 7 (strongly agree). It has 12 items in total, and the total score reflects the total social support felt by the individual. The scale has three dimensions: family support, friends support, and others support. The overall score between 12–36 is a low support status; 37–60 is a medium support status; 61–84 is a high support status; the higher the overall score is, the higher the individual's social support people have (*Ye et al., 2022*). In this study, the Cronbach's alpha for the scale was 0.930.

### Interpersonal sensitivity

The Symptom Check List-90 (SCL-90) is used to assess respondents' interpersonal sensitivity (*Dang et al., 2020*). The SCL-90 invented by Derogatis can be used to assess the intensity of self-reported symptoms, and the scale contains several subscales. The Interpersonal Sensitivity Subscale (nine items in total), one of the subscales, is used to assess the intensity of interpersonal sensitivity among students at Chongqing Medical University. Responses to items are measured on a 5-point Likert scale (0 = none 4 = critical), with a total score ranging from 0–36. Subscale score greater than 2 indicates a psychological abnormality (*Cai et al., 2020*). The Cronbach's alpha measurement for this subscale is 0.855, which is more than 0.8 with high reliability.

### Depression

The Patient Health Questionnaire (PHQ-9) is used to measure depression (*Kroenke, Spitzer & Williams, 2001*). PHQ-9 consists of nine items for depression self-assessment. The scale was widely used because of its efficiency and convenience. It was rated on a 4-point Likert scale with a score ranging from 0 to 3 and a total score range of 0–27 on the scale. The cut-offs have been proposed as 0–4, 5–9, 10–14, 15–19, and 20–27 for no, mild, moderate, moderately severe, and severe depression respectively (*Furukawa et al., 2021*). The Cronbach's alpha for this component in this study was 0.852.

## Statistical analysis

This study used the Statistical Package for the Social Sciences (SPSS) version 25.0 (IBM Corp., Armonk, NY, USA) to analyze the data. Continuous variables are represented using mean and standard deviation (M ± SD), and categorical variables are expressed as frequency and percentage (n(%)). Normality tests and uniformity of variance tests are performed for the measurement of AA, IS, SS, and depression scores under different basic demographic characteristics. Further, a t-test on two independent samples or variance homogeneity is performed to determine whether there was a difference in the mean values of AA, IS, SS, and depression between different subgroups when all samples were satisfied. Pearson's correlation was used to analyze the correlations between the appearance anxiety, social support, interpersonal sensitivity, and depression. To further understand the relationship between the above variables, a serial multiple mediation model (model 6) was performed using the PROCESS macro 3.5 package provided by Hayes (*Xian et al., 2023*). Gender and age were used as covariates, appearance anxiety as the independent variable

(X), interpersonal sensitivity and social support as mediating variables (M1, M2), and depression as the dependent variable (Y). After the model was built, the mediation effects were tested using a bootstrap (5,000 bootstrap samples) based on 95% confidence intervals. If the 95% confidence interval for the mediating effect does not include zero, the mediating effect will be significant at the 0.05 level. The model of this study was tested to be significant at the 0.05 level.

# RESULTS

## Basic demographic variables

All participants were asked to complete a questionnaire that included gender, age, grade, height, weight, whether they were an only child, whether their home address was town or village, monthly household income, GPA for the previous school year, and disposable income. Specific issues and analysis results are shown in Table 1. As shown in Table 1, there are significant differences in the mean of AA scores across grade levels and BMI index (F = 3.402 $P = 0.009$; F = 5.752 $P = 0.003$). IS scores varied significantly by grade levels, home address, and ranking of results (F = 2.738, $P = 0.028$; F = 2.113 $P = 0.035$; F = −2.426 $P = 0.016$). The average scores of SS varied significantly across gender, grade, BMI, only a child, home address, monthly income, ranking of results, single and discretionary income (F = 2.228 $P = 0.026$; F = 2.682 $P = 0.031$; F = 3.367 $P = 0.001$; F = −3.004 $P = 0.003$; F = −5.289 $P < 0.000$; F = 2.801 $P = 0.005$; F = 1.985 $P = 0.048$; F = −3.584 $P < 0.000$). The mean of Depression scores varied with grade levels and ranking of results (F = 2.679 $P = 0.031$; F = −2.885 $P = 0.004$).

## Correlation analysis

The correlation matrix of key study variables is presented in Table 2. IS and AA were positively correlated (r = 0.568, $P < 0.01$); AA and IS were negatively correlated with SS (AA: r = −0.323, $P < 0.01$; IS: r = −0.319, $P < 0.01$); AA and IS were positively correlated with depression (AA: r = 0.438, $P < 0.01$; IS: r = 0.534, $P < 0.01$), and SS was negatively correlated with depression (r = −0.344, $P < 0.01$).

## Multiple mediation analyses of the hypothesized model

A multiple mediation analysis was conducted to explore the mediation effects of IS and SS in a college student population. Control variables included gender, age, BMI, GPA, being an only child or not, and home address. AA and depression were entered as independent and dependent variables respectively. The proposed mediators were IS and SS. Results of the analysis (Table 3) showed that AA was positively correlated with depression (Index = 0.168, $P < 0.001$). Secondly, AA was positively correlated with IS (Index = 0.575, $P < 0.001$) and negatively correlated with SS (Index = −0.234, $P < 0.001$). IS was negatively correlated with SS (Index = −0.179, $P < 0.001$). In addition, IS was positively correlated with depression (Index = 0.384, $P < 0.001$) and SS and depression were negatively correlated (Index = −0.161, $P < 0.001$).

Table 1 Differences in appearance anxiety, interpersonal sensitivity, social support, and depression under different subgroups of basic demographic characteristics.

| Variable | n(%) | Appearance anxiety | t/F(p) | Interpersonal sensitivity | t/F(p) | Social support | t/F(p) | Depression | t/F(p) |
|---|---|---|---|---|---|---|---|---|---|
| Gender | | | −0.150 (0.880) | | 0.424 (0.672) | | 2.228 (0.026) | | −0.701 (0.483) |
| Male | 231 (31.9) | 39.98 ± 8.76 | | 12.48 ± 5.71 | | 58.91 ± 11.22 | | 6.68 ± 4.17 | |
| Female | 493 (68.1) | 39.87 ± 8.64 | | 12.68 ± 5.88 | | 60.98 ± 11.82 | | 6.45 ± 4.18 | |
| Age | | | | 19.8 ± 2.02 | | | | | |
| Grade | | | 3.402 (0.009) | | 2.738 (0.028) | | 2.682 (0.031) | | 2.679 (0.031) |
| Freshman | 256 (35.4) | 40.34 ± 8.64 | | 12.75 ± 5.87 | | 61.87 ± 11.70 | | 6.15 ± 4.14 | |
| Sophomore | 236 (32.6) | 40.47 ± 8.58 | | 13.27 ± 5.87 | | 59.67 ± 11.98 | | 6.70 ± 3.97 | |
| Junior | 167 (23.1) | 39.14 ± 8.90 | | 11.89 ± 5.86 | | 59.70 ± 10.90 | | 7.11 ± 4.68 | |
| Senior | 53 (7.3) | 39.57 ± 8.29 | | 12.19 ± 3.56 | | 57.06 ± 11.72 | | 6.26 ± 3.56 | |
| Above | 12 (1.7) | 31.83 ± 5.27 | | 8.92 ± 3.06 | | 63.25 ± 11.19 | | 3.92 ± 2.75 | |
| BMI | | | 5.752 (0.003) | | 0.739 (0.478) | | 0.036 (0.964) | | 0.308 (0.735) |
| <18.5 | 136 (18.8) | 38.20 ± 8.71 | | 12.15 ± 5.17 | | 60.25 ± 11.95 | | 6.29 ± 4.30 | |
| 18.5–23.9 | 498 (68.8) | 39.97 ± 8.41 | | 12.79 ± 5.90 | | 60.39 ± 11.85 | | 6.55 ± 4.16 | |
| >23.9 | 90 (12.4) | 42.16 ± 9.57 | | 12.37 ± 6.28 | | 60.04 ± 10.20 | | 6.71 ± 4.14 | |
| The only child | | | 0.611 (0.542) | | −0.459 (0.646) | | 3.367 (0.001) | | −0.312 (0.755) |
| Yes | 290 (40.1) | 40.15 ± 8.84 | | 12.49 ± 5.84 | | 62.09 ± 11.92 | | 6.47 ± 4.09 | |
| No | 434 (59.9) | 39.75 ± 8.56 | | 12.70 ± 5.82 | | 59.14 ± 11.35 | | 6.56 ± 4.24 | |
| Home address | | | −0.185 (0.853) | | 2.113 (0.035) | | −3.004 (0.003) | | 1.185 (0.237) |
| Township | 244 (33.7) | 39.82 ± 7.71 | | 13.25 ± 5.67 | | 58.50 ± 10.67 | | 6.78 ± 4.28 | |
| City | 480 (66.3) | 39.95 ± 9.13 | | 12.29 ± 5.87 | | 61.24 ± 12.04 | | 6.39 ± 4.13 | |
| Monthly income | | | 0.218 (0.827) | | 1.328 (0.185) | | −5.289 (<0.000) | | 1.532 (0.126) |
| ≤5,000 | 265 (36.6) | 40.00 ± 8.52 | | 12.99 ± 5.83 | | 57.36 ± 11.37 | | 6.84 ± 4.29 | |
| >5,000 | 459 (63.4) | 39.85 ± 8.77 | | 12.40 ± 5.81 | | 62.03 ± 11.50 | | 6.34 ± 4.11 | |
| Ranking of results | | | −0.677 (0.499) | | −2.426 (0.016) | | 2.801 (0.005) | | −2.885 (0.004) |
| top 50% | 531 (73.3) | 39.78 ± 8.27 | | 12.30 ± 5.69 | | 61.05 ± 11.22 | | 6.26 ± 3.92 | |
| Post 50% | 193 (26.7) | 40.27 ± 9.71 | | 13.48 ± 6.11 | | 58.32 ± 12.62 | | 7.26 ± 4.75 | |
| Single | | | 1.696 (0.090) | | 1.353 (0.176) | | 1.985 (0.048) | | −0.569 (0.570) |
| Yes | 570 (78.7) | 40.19 ± 8.66 | | 12.77 ± 5.83 | | 60.77 ± 11.40 | | 6.48 ± 4.16 | |
| No | 154 (21.3) | 38.86 ± 8.68 | | 12.05 ± 5.78 | | 58.67 ± 12.50 | | 6.69 ± 4.25 | |
| Discretionary income | | | −0.804 (0.422) | | 0.799 (0.424) | | −3.584 (<0.000) | | 0.245 (0.806) |
| <1,500 | 370 (51.1) | 39.65 ± 8.54 | | 12.78 ± 6.05 | | 58.81 ± 11.29 | | 6.56 ± 4.18 | |
| >1,500 | 354 (48.9) | 40.17 ± 8.81 | | 12.44 ± 5.58 | | 61.90 ± 11.85 | | 6.49 ± 4.18 | |

## Bootstrap test of mediators

The mediation path model is presented in Fig. 1. To test the significance of the mediating effect of IS and SS, a bootstrap estimation procedure with 5,000 bootstrap samples is performed. The total effect, direct effect, and indirect effect are presented in Table 4. The path coefficients for the 95% CI of the paths do not include 0 which means that the path coefficient of this path is significantly true at the level of 0.05. As is shown in Table 4, the significance of the direct effect of AA on depression (Effect = 0.081, 95% CI

**Table 2 Correlation analysis of appearance anxiety, interpersonal sensitivity, social support, and depression.**

| Variable | M ± SD | ① | ② | ③ | ④ |
|---|---|---|---|---|---|
| ① Appearance anxiety | 39.91 ± 8.67 | 1 | | | |
| ② Interpersonal sensitivity | 12.61 ± 5.82 | 0.568** | 1 | | |
| ③ Social support | 60.32 ± 11.66 | −0.323** | −0.319** | 1 | |
| ④ Depression | 6.5 ± 4.18 | 0.438** | 0.534** | −0.344** | 1 |

Note:
** $P < 0.01$.

**Table 3 Regression analysis of appearance anxiety, interpersonal sensitivity, social support, and depression.**

| Regression model | | Model fit index | | | | Significance of regression coefficients |
|---|---|---|---|---|---|---|
| Outcome variables | Predictive variables | R | $R^2$ | F | β | t |
| Interpersonal sensitivity | Appearance anxiety | 0.582 | 0.338 | 52.310 | 0.575 | 18.648*** |
| Social support | Appearance anxiety | 0.417 | 0.174 | 18.825 | −0.234 | −5.569*** |
| | Interpersonal sensitivity | | | | −0.179 | −4.289*** |
| Depression | Appearance anxiety | 0.583 | 0.340 | 40.790 | 0.168 | 4.370*** |
| | Interpersonal sensitivity | | | | 0.384 | 10.142*** |
| | Social support | | | | −0.161 | −4.801*** |

Notes:
*** $P < 0.001$.
β is the normalization factor.

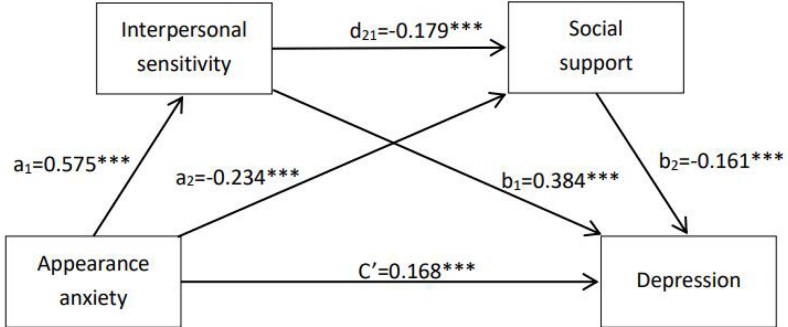

**Figure 1 Generalized linear model serial mediation.** *** $P < 0.001$.

[0.045–0.117]) remained when the mediators (IS and SS) were included in the model. AA was found to indirectly affect depression psychological stress through three significant mediation pathways: (1) IS (B = 0.106, 95% CI [0.082–0.132]), which accounted for 49.77% of the total effect, (2) SS (B = 0.018, 95% CI [0.008–0.031]), which accounted for 8.45% of the total effect, and (3) IS and SS (B = 0.008, 95% CI [0.003–0.014]), which accounted for 3.76% of the total effect. The total mediating effect was 61.97%.

**Table 4 Significance test for mediating effects of appearance anxiety, interpersonal sensitivity, social support, and depression.**

| | Effect | BootSE | BootLLCI | BootULCI | Percentage of total effect |
|---|---|---|---|---|---|
| Total effect | 0.213 | 0.016 | 0.181 | 0.245 | 100% |
| Direct effect | 0.081 | 0.019 | 0.045 | 0.117 | 38.03% |
| Total indirect effect | 0.132 | 0.015 | 0.105 | 0.162 | 61.97% |
| Appearance anxiety → Interpersonal sensitivity → Depression | 0.106 | 0.013 | 0.082 | 0.132 | 49.77% |
| Appearance anxiety → Social support → Depression | 0.018 | 0.006 | 0.008 | 0.031 | 8.45% |
| Appearance anxiety → Interpersonal sensitivity → Social support → Depression | 0.008 | 0.003 | 0.003 | 0.014 | 3.76% |

# DISCUSSION

AA has long been a hot topic in young adults. At the stage of physical and mental maturity, teenagers are delicate and sensitive, and they begin to care about their appearance, but external pressure on their appearance leads to negative emotions. Liao's research shows that 78.8% of medical students in China are concerned about their appearance. It shows that the problem of AA is very serious among medical students, and high AA can lead to social anxiety and depression. Medical students' focus on weight and figures may be related to AA (*Liao et al., 2010*). This study is the first to explore the effects of IS and SS on adolescents' AA and depression in a sample of 724 at Chongqing Medical University in Chongqing, China, and to investigate the effects of SS and IS as mediating variables on AA and depression mechanisms. The results of this study will contribute to the early detection and timely intervention of depression among medical college students and contribute to the healthy physical and mental growth of medical college students studying in medical schools. Moreover, this research can also make up for the research gap in this area at home and abroad in recent years, to have a certain impact on the prevention of mental illness of college students. AA scores vary by grade and BMI, which is possible because contemporary society promotes the concept of "thinness" as beauty (*Liao et al., 2010*). It leads students to pay a lot of attention to their BMI, but further research is needed on the impact of grade level. As mentioned in previous articles, the increasing of age is likely to lead to increased levels of AA (*Carrard et al., 2021*). IS varies by grade, home address, and grades. Freshmen may be at higher risk of IS because they have just entered a new environment and are mentally less mature than older students (*Xu et al., 2022*). SS varies by gender, grade, BMI, only a child, home address, monthly income, ranking of results, single and discretionary income, and depression by grade and test score. Especially students who are undergoing graduate exams in their senior year are more likely to suffer from depression (*Chen et al., 2022*).

This study found that AA positively and significantly predicted the onset of depression in a medical college student population, which is consistent with the findings of a study exploring the relationship between appearance and psychological distress (*Mirza et al., 2021*). It is undeniable that appearance is the most intuitive "first face" in human interaction. People can use make-up to strengthen their confidence in their appearances, but it is more important to establish the correct concept of appearance and lessen the

chance of AA. Previous studies have found that if they are more sensitive and attentive to their appearance than others, they may take a higher risk of depression (*Mirza et al., 2021*). Depression is the leading cause of illness and disability among adolescents (*Ward et al., 2021*). It is not that individuals will never have negative emotions such as depression and anxiety without mental problems at present. Actually, mental health can change. Especially, for example, in the early days of the exam, the number of students with depressive symptoms increases. This is possibly due to excessive psychological stress and challenges. Facing the exam, they have less time to unwind and thus psychological problems are exacerbated (*Chen et al., 2022*). Therefore, more attention should be attached to the mental health of medical students, especially at the examination stage. The right amount of recreational activities and time with their families should be provided to students at this stage to increase SS and reduce their depression. When depression occurs during adolescence, they have a higher rate of recurrence of depression in the future and their quality of life will be seriously affected, which will increase the economic and health burden of society (*Petersen et al., 2022*; *Clayborne, Varin & Colman, 2019*). In this regard, the corresponding relevant departments (especially schools) should attach high importance to the occurrence of depression among college students, conduct early mental health screening in a timely manner, provide additional psychological support to students at risk, and combine early detection with effective intervention in order to reduce the probability of college students suffering from depression and aggravation of the condition (*Mofatteh, 2021*). At the same time, actively guiding students to reduce their external evaluations of others, building a good communication platform, and improving their coping and interpersonal skills can enable students to receive more campus support.

This study shows that IS not only is a mental health problem for medical college students but also it has a great potential to affect depression in college students with appearance anxiety. Medical college students with high IS are overly concerned about the feelings and behaviors of others and fear negative evaluations and rejections from others, therefore they are psychologically fragile and sensitive. Such students are at high risk of losing themselves and becoming depressed in order to avoid rejection and criticism. Previous studies have confirmed that people with IS traits are more likely to develop depression (*Boyce & Parker, 1989*). In this regard, group counseling has been widely used in intervention studies of interpersonal communication and prevention of depression among college students with good results, and previous studies have also confirmed that group sand tray game therapy can effectively reduce IS of college students (*Young et al., 2016*; *Wen et al., 2011*). Schools and other relevant departments can learn from the existing studies and actively carry out activities such as group counseling and group sand tray games to help individuals reduce IS and the occurrence of depression among college students.

This study confirmed the validity of hypothesis 3. SS is an important resilient resource for individuals in social life and a protective factor for college students' physical and mental health (*Gecková et al., 2003*). The result of this study indicating that higher SS can maintain a good emotional experience for individuals, enhance the coping and handle ability of college students when facing AA, and reduce the anxiety and depression caused by

appearance (*Maulik, Eaton & Bradshaw, 2011*). Specifically, SS was negatively correlated with AA, IS, and depression. A decrease in SS can seriously affect people's psychological state, leading to increased levels of anxiety and depression. As a previous study shows, people with higher IS have lower SS, so people with higher SS are better able to face and adapt to relationships (*Montag et al., 2020*). Family and friends should give more love and accompany to youth who are in sensitive periods, encourage and support them when they suffer setbacks, and give them timely help when they encounter difficulties. State authorities should also give high priority to the occurrence and treatment of depression among college students, and help youth receive reliable support and the benefits they deserve. and build a good platform to improve their coping ability and interpersonal skills.

Findings in this study enriched the mediating influences of AA leading to depression: college students with AA faced with higher SS or lower IS would reduce the occurrence of depression. Also, the results found that IS was significantly and negatively correlated with SS (r = −0.319, $P < 0.01$), meaning that among college students with AA, those with lower IS received more SS and thus reduced the incidence of depression. College students, especially medical school students, are the backbone of social construction and development in the future. Although youth is often considered to be the best stage of health, this is by no means a reason to neglect the physical and mental health of college students. All departments should take active measures to attach great importance to the physical and mental health of college students studying in a medical school and draw a blueprint for a healthy and happy future together.

## Limitations

Although all the hypotheses in the study are valid, the limitations of the study must also be acknowledged. Firstly, it is a cross-sectional research method and the causal relationship among AA, IS, SS, and depression is unclear. So the longitudinal studies could be conducted based on this study to further explore the causal relationships between variables. Secondly, the study was conducted about 3 months after the strict prevention and control of COVID-19 had been lifted in China, and because of the wearing of masks, people's anxiety about their appearance may change, which is unknown. Another survey of AA can be conducted several years later and compared it with this study to get conclusions. Thirdly, the population of this study were students of Chongqing Medical University, and due to the special nature of their specialties, they will have a more scientific perception of physiological appearance than other college students. If the findings of this study are to be extended to other specialties or schools, the representative sample needs to be further expanded to reduce the error. In addition, unfortunately, we did not further exclude people with previous or current major psychological problems in our study, which may bias our findings.

## CONCLUSIONS

The study confirmed that the four hypotheses mentioned in the introduction hold. Specifically, lower IS and higher SS can effectively reduce depression caused by AA among Chongqing medical students. It also reconfirmed that AA can directly or indirectly

influence the occurrence of depression in medical college students. In this study, the presence of IS and SS were found to work as mediators regulating the mechanism of action between AA and depression. It helps to clarify the potential mechanisms between AA and depression. The aim is to improve the physical and mental health of medical college students and appeal to society to give high priority to young people. Schools and other relevant departments can learn from the existing studies and actively carry out activities such as group counseling and group sand tray games to help individuals reduce IS and the occurrence of depression among college students studying in medical schools. At the same time, medical college students should also actively communicate with people around them, improve their interpersonal skills, and avoid over-interpreting the feelings and behaviors of others to reduce their psychological burden.

## ACKNOWLEDGEMENTS

We thank every participant for their support of our study.

### Funding

This research was funded by the Humanities and Social Sciences Research Project Fund of Chongqing Municipal Education Commission in 2022, grant number NO.22SKGH052. The funders had no role in study design, data collection and analysis, decision to publish, or preparation of the manuscript.

### Grant Disclosures

The following grant information was disclosed by the authors:
Chongqing Municipal Education Commission: 22SKGH052.

### Competing Interests

The authors declare no conflict of interest.

### Author Contributions

- Xiaobing Xian conceived and designed the experiments, performed the experiments, analyzed the data, prepared figures and/or tables, authored or reviewed drafts of the article, and approved the final draft.
- Tengfei Niu conceived and designed the experiments, authored or reviewed drafts of the article, and approved the final draft.
- Yu Zhang performed the experiments, analyzed the data, authored or reviewed drafts of the article, and approved the final draft.
- Xilin Zhou performed the experiments, analyzed the data, authored or reviewed drafts of the article, and approved the final draft.
- Xinxin Wang performed the experiments, authored or reviewed drafts of the article, and approved the final draft.
- Xin Du performed the experiments, authored or reviewed drafts of the article, and approved the final draft.

- Linhan Qu performed the experiments, prepared figures and/or tables, authored or reviewed drafts of the article, and approved the final draft.
- Binyi Mao performed the experiments, authored or reviewed drafts of the article, and approved the final draft.
- Ying He performed the experiments, authored or reviewed drafts of the article, and approved the final draft.
- Xiyu Chen performed the experiments, authored or reviewed drafts of the article, and approved the final draft.
- Mengliang Ye conceived and designed the experiments, performed the experiments, analyzed the data, prepared figures and/or tables, authored or reviewed drafts of the article, and approved the final draft.

## Data Availability
The raw data are available in the Supplemental File.

## Supplemental Information
Supplemental information for this article can be found online at http://dx.doi.org/10.7717/peerj.17090#supplemental-information.

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
