# Peer review of "The relationship between appearance anxiety and depression among students in a medical university in China: a serial multiple mediation model"

_PeerJ, doi:10.7717/peerj.17090_

## Round 0.1 · original submission · Major Revisions

Dear Authors

Thank you for your submission. Looking at the Reviews, they provide a varied range of comments and suggestions. Given that two of the Reviewers see merit in the paper, I invite you to resubmit the article after addressing Reviewer comments. Please also include a response to the brief criticisms provided by Reviewer 3. That is, explain why the sample was representative and/or why it was included within the study. This could include an acknowledgement of potential limitations and future remedies/extensions.

Reviewer 1 has suggested that you cite specific references. You are welcome to add it/them if you believe they are relevant. However, you are not required to include these citations, and if you do not include them, this will not influence my decision.

Best Wishes
Neil

Reviewer 1 ·

Basic reporting

It is well known that adolescents can be more prone to appearance anxiety and they may be at higher risk for depression. However, few studies have revealed the mechanisms linking appearance anxiety and depression in college students. In this study, the authors aimed to explore the multiple mediating roles of interpersonal sensitivity and social support between appearance anxiety and depression among college students. In total, 724 college students completed questionnaires containing basic demographic characteristics, appearance anxiety scale, interpersonal sensitivity scale, perceived social support scale, and depression scale. The data revealed that appearance anxiety can not only directly cause depression, but also indirectly affect depression through three mediating pathways, including interpersonal sensitivity, perceived social, and interpersonal sensitivity and perceived social support.

This is an interesting study on an important topic, however, there are major comments which need to be addressed before the manuscript is considered further.

I provide detailed comments below to help the authors improve their manuscript.

1. Abstract: Please provide some more basic information on participants, including age, gender, college year, etc.

2. Introduction: Lines 49-51: “The onset of depression can lead to a negative state of mind, poor sleep quality, and health-hazardous behaviors such as smoking, drinking, substance abuse, and poor eating and living habits” Please discuss that broader risk factors, including biological, psychological, social, and academic can affect stress, anxiety and depression among college students. The following paper can be cited:
Risk factors associated with stress, anxiety, and depression among university undergraduate students. AIMS Public Health. 2020 Dec 25;8(1):36-65. doi: 10.3934/publichealth.2021004. PMID: 33575406; PMCID: PMC7870388.

3. Introduction: Line 53: Change “What’s more important” to “More importantly”

4. Introduction: Line 62: Change “proposed: when” to “proposed that when”

5. Introduction: “A cross-sectional study showed that people who had their eyes removed with higher levels of concern about their appearance than others were more likely to suffer from depression”. This is not quite relevant to the current study, unless the authors specifically investigated the effect of disability on appearance anxiety among adolescents.

6. Introduction: Line 89: Change “Further research is needed.” To “Further research is needed to validate and extend these findings.”

Experimental design

I provide specific comments below:

7. Materials and Methods: The recruitment process was quite short. Can the authors please explain how they recruited 737 students over 3 days?

8. Methods: Did any of the students have previous mental health problems? Were such students included in the study? If yes, how this can affect the findings?

9. The inclusion/exclusion criteria are very brief.

Validity of the findings

Specific comments are below:

10. Results: The Results section needs improvement. There is no information on mean age and age range, sex/gender, grade (first grade, second grade, …), GPA, mental health status, comorbidity, social status, family income, BMI, etc of students. A Table should be added and such information should be provided.

11. Results: Multiple mediation analyses of the hypothesized model: Why BMI and GPA were not included? Why only age and gender were included? Please explain.

Additional comments

12. Discussion: “in patients with rheumatic diseases” How is this citation relevant to the current study? Did any of the students have rheumatic disease? If not, please use more relevant references.

13. Is AA more relevant for medical students compared to students studying other subjects, for example because they are more seen by people and they need to behave better and also they need to care about other people’s/patients’ feelings? Please discuss.

14. Discussion: “Depression is the leading cause of illness and disability among adolescents” Please discuss broadly that the mental health of students can change, for example during exam period, and it can recover later on. The following paper can be cited:
a. Chen et al. Mental health status of medical students during postgraduate entrance examination. BMC Psychiatry. 2022 Dec 27;22(1):829. doi: 10.1186/s12888-022-04482-1. PMID: 36575395

15. Discussion: Line 265: This sentence should move to Conclusion: “Schools and other relevant departments can learn from the existing studies and actively carry out activities such as group counseling and group sand tray games to help individuals reduce IS and the occurrence of depression among college students.”

16. Discussion: Line 282: “Relevant departments should give high priority to the occurrence of depression among college…” Please discuss that students with mental health problems need to be identified early and special support should be provided to them to prevent worsening of their mental health problem. This has been suggested in the paper below, which can be cited:
a. Risk factors associated with stress, anxiety, and depression among university undergraduate students. AIMS Public Health. 2020 Dec 25;8(1):36-65. doi: 10.3934/publichealth.2021004. PMID: 33575406; PMCID: PMC7870388.

17. Conclusion: Please move Limitations to before Conclusion.

·

Basic reporting

The paper explores a very important problem, especially in the social media era. To this point, and because it may not be well known to the Western world how social media works in China (due to the restrictions we all hear about), the paper could benefit from some general contextualization.

The sentence starting on line 48 and ending on line 49 with the reference n.º 4 could benefit from the explanation: why do they have an increased risk? There is the reference, but it is much more enriching when the affirmations are supported by more than a reference.

Line 67-69 (reference 16): This sentence could benefit from the explanation: why do they have an increased risk? There is the reference, but it is much more enriching when the affirmations are supported by more than a reference.

The results would be easier to read and understand if organized by hypotheses.

In the discussion, lines 226-229: There may be some important differences between AA in Medical students and other students, for instance, art students. So, this generalization must be carefully rephrased.

Lines 273-275: This sentence does not add relevant information to the study.

The tables are clean and easy to understand, as well as the figure.

Experimental design

The research questions are well-exposed and the methods to explore the hypothesis were well thought off. Future research should explore the subject in students from other fields and schools.

Validity of the findings

I believe the results of this study are of extreme importance and relevance; however as the sample was composed only of medical students and medical students (as health students in general) have been reported to have particular characteristics when compared with students from other fields, the generalization to the university students universe must be made with precaution.

Reviewer 3 ·

Basic reporting

A thorough check and language editing are needed.

Experimental design

Big problems in the samples.

Validity of the findings

The results may not be representative to present the patterns of “appearance anxiety and depression among Chinese college students”.

Additional comments

Please refer to the attached file.

Annotated reviews are not available for download in order to protect the identity of reviewers who chose to remain anonymous.

---

## Round 0.2 · Major Revisions

Please see the comments from Reviewer 3. Could you please address their remaining concerns and comments. While the submission has improved, further amendments are required.

Reviewer 1 ·

Basic reporting

The authors have addressed my comments. Therefore, I recommend acceptance of the manuscript.

Experimental design

The manuscript is well written.

Validity of the findings

The authors have improved their work significantly after the revision.

·

Basic reporting

No comment

Experimental design

No comment

Validity of the findings

No comment

Reviewer 3 ·

Basic reporting

Although some grammar mistakes were revised, there were still many informal sentences. Professional language editing is suggested.
Please see detailed comments in the attached file.

Experimental design

Please see detailed comments in the attached file.

Validity of the findings

Please see detailed comments in the attached file.

Additional comments

Please see detailed comments in the attached file.

Annotated reviews are not available for download in order to protect the identity of reviewers who chose to remain anonymous.

---

## Round 0.3 · Minor Revisions

Please see the comments from the Reviewer, I would like you to address these thoughly and ensure that you check for similar issues throughout the submission.

Reviewer 3 ·

Basic reporting

Language editing is still needed. Mistakes can be seen in many parts, like the following:
1. With invalid 13 invalid samples excluded
2. Hayes' PROCESS macro is used for SPSS. PROCESS macro for SPSS was used.
3. Anxiety about appearance make them; they generates negative emotions, and so on
4. a convenience sampling was used to recruit 737 college students for questionnaire survey at Chongqing Medical University. Posters, applets, QR codes, and links are utilized to disseminate and recruit research subjects. Two tenses?
5. The efficiency and convenience of the scale makes itself widely used?? makes? makes itself widely used?
6. the numbers of students with depressive symptoms increase. numbers?
and many others, I did not list all.

Experimental design

has improved in this version.

Validity of the findings

has improved in this version.

Additional comments

no

---

## Round 0.4 · accepted · Accept

Thank you for the resubmission, The reviewers are now satisfied.

Reviewer 3 ·

Basic reporting

improved

Experimental design

improved

Validity of the findings

improved